# Expectation Particle Belief Propagation

**Thibaut Lienart, Yee Whye Teh, Arnaud Doucet**
Department of Statistics
University of Oxford
Oxford, UK
{lienart,teh,doucet}@stats.ox.ac.uk

## Abstract

We propose an original particle-based implementation of the Loopy Belief Propagation (LPB) algorithm for pairwise Markov Random Fields (MRF) on a continuous state space. The algorithm constructs adaptively efficient proposal distributions approximating the local beliefs at each note of the MRF. This is achieved by considering proposal distributions in the exponential family whose parameters are updated iterately in an Expectation Propagation (EP) framework. The proposed particle scheme provides consistent estimation of the LBP marginals as the number of particles increases. We demonstrate that it provides more accurate results than the Particle Belief Propagation (PBP) algorithm of [1] at a fraction of the computational cost and is additionally more robust empirically. The computational complexity of our algorithm at each iteration is quadratic in the number of particles. We also propose an accelerated implementation with sub-quadratic computational complexity which still provides consistent estimates of the loopy BP marginal distributions and performs almost as well as the original procedure.

## 1 Introduction

Undirected Graphical Models (also known as *Markov Random Fields*) provide a flexible framework to represent networks of random variables and have been used in a large variety of applications in machine learning, statistics, signal processing and related fields [2]. For many applications such as tracking [3, 4], sensor networks [5, 6] or computer vision [7, 8, 9] it can be beneficial to define MRF on continuous state-spaces.

Given a pairwise MRF, we are here interested in computing the marginal distributions at the nodes of the graph. A popular approach to do this is to consider the Loopy Belief Propagation (LBP) algorithm [10, 11, 2]. LBP relies on the transmission of *messages* between nodes. However when dealing with continuous random variables, computing these messages exactly is generally intractable. In practice, one must select a way to tractably represent these messages and a way to update these representations following the LBP algorithm. The *Nonparametric Belief Propagation* (NBP) algorithm [12] represents the messages with mixtures of Gaussians while the *Particle Belief Propagation* (PBP) algorithm [1] uses an importance sampling approach. NBP relies on restrictive integrability conditions and does not offer consistent estimators of the LBP messages. PBP offers a way to circumvent these two issues but the implementation suggested proposes sampling from the estimated beliefs which need not be integrable. Moreover, even when they are integrable, sampling from the estimated beliefs is very expensive computationally. Practically the authors of [1] only sample approximately from those using short MCMC runs, leading to biased estimators.

In our method, we consider a sequence of proposal distributions at each node from which one can sample particles at a given iteration of the LBP algorithm. The messages are then computed using importance sampling. The novelty of the approach is to propose a principled and automated way of designing a sequence of proposals in a tractable exponential family using the Expectation Prop-

agation (EP) framework [13]. The resulting algorithm, which we call *Expectation Particle Belief Propagation* (EPBP), does not suffer from restrictive integrability conditions and sampling is done exactly which implies that we obtain consistent estimators of the LBP messages. The method is empirically shown to yield better approximations to the LBP beliefs than the implementation suggested in [1], at a much reduced computational cost, and than EP.

## 2 Background

### 2.1 Notations

We consider a pairwise MRF, i.e. a distribution over a set of $p$ random variables indexed by a set $V = \{1, \ldots, p\}$, which factorizes according to an undirected graph $G = (V, E)$ with

$$p(x_V) \quad \propto \quad \prod_{u \in V} \psi_u(x_u) \prod_{(u,v) \in E} \psi_{uv}(x_u, x_v). \tag{1}$$

The random variables are assumed to take values on a continuous, possibly unbounded, space $\mathcal{X}$. The positive functions $\psi_u : \mathcal{X} \mapsto \mathbb{R}_+$ and $\psi_{uv} : \mathcal{X} \times \mathcal{X} \mapsto \mathbb{R}_+$ are respectively known as the *node* and *edge potentials*. The aim is to approximate the marginals $p_u(x_u)$ for all $u \in V$. A popular approach is the LBP algorithm discussed earlier. This algorithm is a fixed point iteration scheme yielding approximations called the *beliefs* at each node [10, 2]. When the underlying graph is a tree, the resulting beliefs can be shown to be proportional to the exact marginals. This is not the case in the presence of loops in the graph. However, even in these cases, LBP has been shown to provide good approximations in a wide range of situations [14, 11]. The LBP fixed-point iteration can be written as follows at iteration $t$:

$$\begin{cases} m_{uv}^t(x_v) & = \quad \displaystyle\int \psi_{uv}(x_u, x_v)\psi_u(x_u) \prod_{w \in \Gamma_u \backslash v} m_{wu}^{t-1}(x_u)\mathrm{d}x_u \\ B_u^t(x_u) & = \quad \psi_u(x_u) \displaystyle\prod_{w \in \Gamma_u} m_{wu}^t(x_u) \end{cases}, \tag{2}$$

where $\Gamma_u$ denotes the neighborhood of $u$ i.e., the set of nodes $\{w \mid (w, u) \in E\}$, $m_{uv}$ is known as the *message* from node $u$ to node $v$ and $B_u$ is the belief at node $u$.

### 2.2 Related work

The crux of any generic implementation of LBP for continuous state spaces is to select a way to represent the messages and design an appropriate method to compute/approximate the message update.

In *Nonparametric BP* (NBP) [12], the messages are represented by mixtures of Gaussians. In theory, computing the product of such messages can be done analytically but in practice this is impractical due to the exponential growth in the number of terms to consider. To circumvent this issue, the authors suggest an importance sampling approach targeting the beliefs and fitting mixtures of Gaussians to the resulting weighted particles. The computation of the update (2) is then always done over a constant number of terms.

A restriction of "vanilla" Nonparametric BP is that the messages must be finitely integrable for the message representation to make sense. This is the case if the following two conditions hold:

$$\sup_{x_v} \int \psi_{uv}(x_u, x_v)\mathrm{d}x_u \quad < \quad \infty, \text{ and } \int \psi_u(x_u)\mathrm{d}x_u \quad < \quad \infty. \tag{3}$$

These conditions do however not hold in a number of important cases as acknowledged in [3]. For instance, the potential $\psi_u(x_u)$ is usually proportional to a likelihood of the form $p(y_u|x_u)$ which need not be integrable in $x_u$. Similarly, in imaging applications for example, the edge potential can encode similarity between pixels which also need not verify the integrability condition as in [15]. Further, NBP does not offer consistent estimators of the LBP messages.

*Particle BP* (PBP) [1] offers a way to overcome the shortcomings of NBP: the authors also consider importance sampling to tackle the update of the messages but without fitting a mixture of Gaussians.

For a chosen proposal distribution $q_u$ on node $u$ and a draw of $N$ particles $\{x_u^{(i)}\}_{i=1}^N \sim q_u(x_u)$, the messages are represented as mixtures:

$$\widehat{m}_{uv}^{\mathrm{PBP}}(x_v) := \sum_{i=1}^N \omega_{uv}^{(i)} \psi_{uv}(x_u^{(i)}, x_v), \quad \text{with} \quad \omega_{uv}^{(i)} := \frac{1}{N} \frac{\psi_u(x_u^{(i)})}{q_u(x_u^{(i)})} \prod_{w \in \Gamma_u \setminus v} \widehat{m}_{wu}^{\mathrm{PBP}}(x_u^{(i)}). \tag{4}$$

This algorithm has the advantage that it does not require the conditions (3) to hold. The authors suggest two possible choices of sampling distributions: sampling from the local potential $\psi_u$, or sampling from the current belief estimate. The first case is only valid if $\psi_u$ is integrable w.r.t. $x_u$ which, as we have mentioned earlier, might not be the case in general and the second case implies sampling from a distribution of the form

$$\widehat{B}_u^{\mathrm{PBP}}(x_u) \ \propto \ \psi_u(x_u) \prod_{w \in \Gamma_u} \widehat{m}_{wu}^{\mathrm{PBP}}(x_u) \tag{5}$$

which is a product of mixtures. As in NBP, naïve sampling of the proposal has complexity $\mathcal{O}(N^{|\Gamma_u|})$ and is thus in general too expensive to consider. Alternatively, as the authors suggest, one can run a short MCMC simulation targeting it which reduces the complexity to order $\mathcal{O}(|\Gamma_u|N^2)$ since the cost of each iteration, which requires evaluating $\widehat{B}_u^{\mathrm{PBP}}$ point-wise, is of order $\mathcal{O}(|\Gamma_u|N)$, and we need $\mathcal{O}(N)$ iterations of the MCMC simulation. The issue with this approach is that it is still computationally expensive, and it is unclear how many iterations are necessary to get $N$ good samples.

### 2.3 Our contribution

In this paper, we consider the general context where the edge and node-potentials might be non-normalizable and non-Gaussian. Our proposed method is based on PBP, as PBP is theoretically better suited than NBP since, as discussed earlier, it does not require the conditions (3) to hold, and, provided that one samples from the proposals exactly, it yields consistent estimators of the LBP messages while NBP does not. Further, the development of our method also formally shows that considering proposals close to the beliefs, as suggested by [1], is a good idea. Our core observation is that since sampling from a proposal of the form (5) using MCMC simulation is very expensive, we should consider using a more tractable proposal distribution instead. However it is important that the proposal distribution is constructed adaptively, taking into account evidence collected through the message passing itself, and we propose to achieve this by using proposal distributions lying in a tractable exponential family, and adapted using the Expectation Propagation (EP) framework [13].

## 3 Expectation Particle Belief Propagation

Our aim is to address the issue of selecting the proposals in the PBP algorithm. We suggest using exponential family distributions as the proposals on a node for computational efficiency reasons, with parameters chosen adaptively based on current estimates of beliefs and EP. Each step of our algorithm involves both a projection onto the exponential family as in EP, as well as a particle approximation of the LBP message, hence we will refer to our method as *Expectation Particle Belief Propagation* or EPBP for short.

For each pair of adjacent nodes $u$ and $v$, we will use $m_{uv}(x_v)$ to denote the exact (but unavailable) LBP message from $u$ to $v$, $\widehat{m}_{uv}(x_v)$ to denote the particle approximation of $m_{uv}$, and $\eta_{uv}$ an exponential family projection of $\widehat{m}_{uv}$. In addition, let $\eta_{\circ u}$ denote an exponential family projection of the node potential $\psi_u$. We will consider approximations consisting of $N$ particles. In the following, we will derive the form of our particle approximated message $\widehat{m}_{uv}(x_v)$, along with the choice of the proposal distribution $q_u(x_u)$ used to construct $\widehat{m}_{uv}$. Our starting point is the edge-wise belief over $x_u$ and $x_v$, given the incoming particle approximated messages,

$$\widehat{B}_{uv}(x_u, x_v) \propto \psi_{uv}(x_u, x_v) \psi_u(x_u) \psi_v(x_v) \prod_{w \in \Gamma_u \setminus v} \widehat{m}_{wu}(x_u) \prod_{\nu \in \Gamma_v \setminus u} \widehat{m}_{\nu v}(x_v). \tag{6}$$

The exact LBP message $m_{uv}(x_v)$ can be derived by computing the marginal distribution $\widehat{B}_{uv}(x_v)$, and constructing $m_{uv}(x_v)$ such that

$$\widehat{B}_{uv}(x_v) \propto m_{uv}(x_v) \widehat{M}_{vu}(x_v), \tag{7}$$

where $\widehat{M}_{vu}(x_v) = \psi_v(x_v) \prod_{\nu \in \Gamma_v \setminus u} \widehat{m}_{\nu v}(x_v)$ is the (particle approximated) pre-message from $v$ to $u$. It is easy to see that the resulting message is as expected,

$$m_{uv}(x_v) \propto \int \psi_{uv}(x_u, x_v) \psi_u(x_u) \prod_{w \in \Gamma_u \setminus v} \widehat{m}_{wu}(x_u) dx_u. \qquad (8)$$

Since the above exact LBP belief and message are intractable in our scenario of interest, the idea is to use an importance sampler targeting $\widehat{B}_{uv}(x_u, x_v)$ instead. Consider a proposal distribution of the form $q_u(x_u) q_v(x_v)$. Since $x_u$ and $x_v$ are independent under the proposal, we can draw $N$ independent samples, say $\{x_u^{(i)}\}_{i=1}^N$ and $\{x_v^{(j)}\}_{j=1}^N$, from $q_u$ and $q_v$ respectively. We can then approximate the belief using a $N \times N$ cross product of the particles,

$$\widehat{B}_{uv}(x_u, x_v) \approx \frac{1}{N^2} \sum_{i,j=1}^N \frac{\widehat{B}_{uv}(x_u^{(i)}, x_v^{(j)})}{q_u(x_u^{(i)}) q_v(x_v^{(j)})} \delta_{(x_u^{(i)}, x_v^{(j)})}(x_u, x_v) \qquad (9)$$

$$\propto \frac{1}{N^2} \sum_{i,j=1}^N \frac{\psi_{uv}(x_u^{(i)}, x_v^{(j)}) \psi_u(x_u^{(i)}) \widehat{M}_{vu}(x_v^{(j)}) \prod_{w \in \Gamma_u \setminus v} \widehat{m}_{wu}(x_u^{(i)})}{q_u(x_u^{(i)}) q_v(x_v^{(j)})} \delta_{(x_u^{(i)}, x_v^{(j)})}(x_u, x_v)$$

Marginalizing onto $x_v$, we have the following particle approximation to $\widehat{B}_{uv}(x_v)$,

$$\widehat{B}_{uv}(x_v) \approx \frac{1}{N} \sum_{j=1}^N \frac{\widehat{m}_{uv}(x_v^{(j)}) \widehat{M}_{vu}(x_v^{(j)})}{q_v(x_v^{(j)})} \delta_{x_v^{(j)}}(x_v) \qquad (10)$$

where the particle approximated message $\widehat{m}_{uv}(x_v)$ from $u$ to $v$ has the form of the message representation in the PBP algorithm (4).

To determine sensible proposal distributions, we can find $q_u$ and $q_v$ that are close to the target $\widehat{B}_{uv}$. Using the KL divergence $\mathrm{KL}(\widehat{B}_{uv} \| q_u q_v)$ as the measure of closeness, the optimal $q_u$ required for the $u$ to $v$ message is the node belief,

$$\widehat{B}_{uv}(x_u) \propto \psi_u(x_u) \prod_{w \in \Gamma_u} \widehat{m}_{wu}(x_u) \qquad (11)$$

thus supporting the claim in [1] that a good proposal to use is the current estimate of the node belief. As pointed out in Section 2, it is computationally inefficient to use the particle approximated node belief as the proposal distribution. An idea is to use a tractable exponential family distribution for $q_u$ instead, say

$$q_u(x_u) \propto \eta_{\circ u}(x_u) \prod_{w \in \Gamma_u} \eta_{wu}(x_u) \qquad (12)$$

where $\eta_{\circ u}$ and $\eta_{wu}$ are exponential family approximations of $\psi_u$ and $\widehat{m}_{wu}$ respectively. In Section 4 we use a Gaussian family, but we are not limited to this. Using the framework of expectation propogation (EP) [13], we can iteratively find good exponential family approximations as follows. For each $w \in \Gamma_u$, to update the $\eta_{wu}$, we form the *cavity distribution* $q_u^{\setminus w} \propto q_u / \eta_{wu}$ and the corresponding *tilted distribution* $\widehat{m}_{wu} q_u^{\setminus w}$. The updated $\eta_{wu}^+$ is the exponential family factor minimising the KL divergence,

$$\eta_{wu}^+ = \arg \min_{\eta \in \text{exp.fam.}} \mathrm{KL}\left[ \widehat{m}_{wu}(x_u) q_u^{\setminus w}(x_u) \,\middle\|\, \eta(x_u) q_u^{\setminus w}(x_u) \right]. \qquad (13)$$

Geometrically, the update projects the tilted distribution onto the exponential family manifold. The optimal solution requires computing the moments of the tilted distribution through numerical quadrature, and selecting $\eta_{wu}$ so that $\eta_{wu} q_u^{\setminus w}$ matches the moments of the tilted distribution. In our scenario the moment computation can be performed crudely on a small number of evaluation points since it only concerns the updating of the importance sampling proposal. If an optimal $\eta$ in the exponential family does not exist, e.g. in the Gaussian case that the optimal $\eta$ has a negative variance, we simply revert $\eta_{wu}$ to its previous value [13]. An analogous update is used for $\eta_{\circ u}$.

In the above derivation, the expectation propagation steps for each incoming message into $u$ and for the node potential are performed first, to fit the proposal to the current estimated belief at $u$, before

it is used to draw $N$ particles, which can then be used to form the particle approximated messages from $u$ to each of its neighbours. Alternatively, once each particle approximated message $\widehat{m}_{uv}(x_v)$ is formed, we can update its exponential family projection $\eta_{uv}(x_v)$ immediately. This alternative scheme is described in Algorithm 1.

---

**Algorithm 1** Node update

---

1: sample $\{x_u^{(i)}\} \sim q_u(\cdot)$
2: compute $\widehat{B}_u(x_u^{(i)}) = \psi_u(x_u^{(i)}) \prod_{w \in \Gamma_u} \widehat{m}_{wu}(x_u^{(i)})$
3: **for** $v \in \Gamma_u$ **do**
4:     compute $\widehat{M}_{uv}(x_u^{(i)}) := \widehat{B}_u(x_u^{(i)}) / \widehat{m}_{vu}(x_u^{(i)})$
5:     compute the normalized weights $w_{uv}^{(i)} \propto \widehat{M}_{uv}(x_u^{(i)}) / q_u(x_u^{(i)})$
6:     update the estimator of the outgoing message $\widehat{m}_{uv}(x_v) = \sum_{i=1}^{N} w_{uv}^{(i)} \psi_{uv}(x_u^{(i)}, x_v)$
7:     compute the cavity distribution $q_v^{\backslash\circ} \propto q_v / \eta_{\circ v}$, get $\eta_{\circ v}^+$ in the exponential family such that $\eta_{\circ v}^+ q_v^{\backslash\circ}$ approximates $\psi_v q_v^{\backslash\circ}$, update $q_v \propto \eta_{\circ v}^+$ and let $\eta_{\circ v} \leftarrow \eta_{\circ v}^+$
8:     compute the cavity distribution $q_v^{\backslash u} \propto q_v / \eta_{uv}$, get $\eta_{uv}^+$ in the exponential family such that $\eta_{uv}^+ q_v^{\backslash u}$ approximates $\widehat{m}_{uv} q_v^{\backslash u}$, update $q_v \propto \eta_{uv}^+$ and let $\eta_{uv} \leftarrow \eta_{uv}^+$
9: **end for**

---

### 3.1 Computational complexity and sub-quadratic implementation

Each EP projection step costs $\mathcal{O}(N)$ computations since the message $\widehat{m}_{wu}$ is a mixture of $N$ components (see (4)). Drawing $N$ particles from the exponential family proposal $q_u$ costs $\mathcal{O}(N)$. The step with highest computational complexity is in evaluating the particle weights in (4). Indeed, evaluating the mixture representation of a message on a single point is $\mathcal{O}(N)$, and we need to compute this for each of $N$ particles. Similarly, evaluating the estimator of the belief on $N$ sampling points at node $u$ requires $\mathcal{O}(|\Gamma_u|N^2)$. This can be reduced since the algorithm still provides consistent estimators if we consider the evaluation of unbiased estimators of the messages instead. Since the messages have the form $\widehat{m}_{uv}(x_v) = \sum_{i=1}^{N} w_{uv}^i \psi_{uv}^i(x_v)$, we can follow a method presented in [16] where one draws $M$ indices $\{i_\ell^\star\}_{\ell=1}^M$ from a multinomial with weights $\{w_{uv}^i\}_{i=1}^N$ and evaluates the corresponding $M$ components $\psi_{uv}^{i_\ell^\star}$. This reduces the cost of the evaluation of the beliefs to $\mathcal{O}(|\Gamma_u|MN)$ which leads to an overall sub-quadratic complexity if $M$ is $o(N)$. We show in the next section how it compares to the quadratic implementation when $M = \mathcal{O}(\log N)$.

## 4 Experiments

We investigate the performance of our method on MRFs for two simple graphs. This allows us to compare the performance of EPBP to the performance of PBP in depth. We also illustrate the behavior of the sub-quadratic version of EPBP. Finally we show that EPBP provides good results in a simple denoising application.

### 4.1 Comparison with PBP

We start by comparing EPBP to PBP as implemented by Ihler et al. on a $3 \times 3$ grid (figure 1) with random variables taking values on $\mathbb{R}$. The node and edge potentials are selected such that the marginals are multimodal, non-Gaussian and skewed with

$$\begin{cases} \psi_u(x_u) & = & \alpha_1 \mathcal{N}(x_u - y_u; -2, 1) + \alpha_2 \mathcal{G}(x_u - y_u; 2, 1.3) \\ \psi_{uv}(x_u, x_v) & = & \mathcal{L}(x_u - x_v; 0, 2) \end{cases} , \tag{14}$$

where $y_u$ denotes the observation at node $u$, $\mathcal{N}(x; \mu, \sigma) \propto \exp(-x^2/2\sigma^2)$ (density of a Normal distribution), $\mathcal{G}(x; \mu, \beta) \propto \exp(-(x-\mu)/\beta + \exp(-(x-\mu)/\beta))$ (density of a Gumbel distribution) and $\mathcal{L}(x; \mu, \beta) \propto \exp(-|x-\mu|/\beta)$ (density of a Laplace distribution). The parameters $\alpha_1$ and $\alpha_2$ are respectively set to $0.6$ and $0.4$. We compare the two methods after 20 LBP iterations.[1]

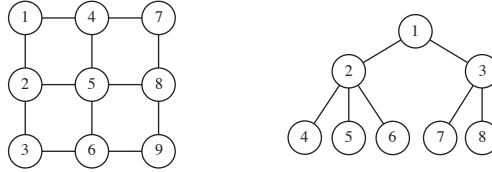

Figure 1: Illustration of the grid (left) and tree (right) graphs used in the experiments.

PBP as presented in [1] is implemented using the same parameters than those in an implementation code provided by the authors: the proposal on each node is the last estimated belief and sampled with a 20-step MCMC chain, the MH proposal is a normal distribution. For EPBP, the approximation of the messages are Gaussians. The ground truth is approximated by running LBP on a deterministic equally spaced mesh with 200 points. All simulations were run with Julia on a Mac with 2.5 GHz Intel Core i5 processor, our code is available online.[2]

Figure 2 compares the performances of both methods. The error is computed as the mean $L^1$ error over all nodes between the estimated beliefs and the ground truth evaluated over the same deterministic mesh. One can observe that not only does PBP perform worse than EPBP but also that the error plateaus with increasing number of samples. This is because the secondampling within PBP is done approximately and hence the consistency of the estimators is lost. The speed-up offered by EPBP is very substantial (figure 4 left). Hence, although it would be possible to use more MCMC (Metropolis-Hastings) iterations within PBP to improve its performance, it would make the method prohibitively expensive to use. Note that for EPBP, one observes the usual $1/\sqrt{N}$ convergence of particle methods.

Figure 3 compares the estimator of the beliefs obtained by the two methods for three arbitrarily picked nodes (node 1, 5 and 9 as illustrated on figure 1). The figure also illustrates the last proposals constructed with our approach and one notices that their supports match closely the support of the true beliefs. Figure 4 left illustrates how the estimated beliefs converge as compared to the true beliefs with increasing number of iterations. One can observe that PBP converges more slowly and that the results display more variability which might be due to the MCMC runs being too short.

We repeated the experiments on a tree with 8 nodes (figure 1 right) where we know that, at convergence, the beliefs computed using BP are proportional to the true marginals. The node and edge potentials are again picked such that the marginals are multimodal with

$$\begin{cases} \psi_u(x_u) & = & \alpha_1 \mathcal{N}(x_u - y_u; -2, 1) + \alpha_2 \mathcal{N}(x_u - y_u; 1, 0.5) \\ \psi_{uv}(x_u, x_v) & = & \mathcal{L}(x_u - x_v; 0, 1) \end{cases} , \tag{15}$$

with $\alpha_1 = 0.3$ and $\alpha_2 = 0.7$. On this example, we also show how "pure EP" with normal distributions performs. We also try using the distributions obtained with EP as proposals for PBP (referred to as "PBP after EP" in figures). Both methods underperform compared to EPBP as illustrated visually in Figure 5. In particular one can observe in Figure 3 that "PBP after EP" converges slower than EPBP with increasing number of samples.

## 4.2 Sub-quadratic implementation and denoising application

As outlined in Section 3.1, in the implementation of EPBP one can use an unbiased estimator of the edge weights based on a draw of $M$ components from a multinomial. The complexity of the resulting algorithm is $\mathcal{O}(MN)$. We apply this method to the $3 \times 3$ grid example in the case where $M$ is picked to be roughly of order $\log(N)$: i.e., for $N = \{10, 20, 50, 100, 200, 500\}$, we pick $M = \{5, 6, 8, 10, 11, 13\}$. The results are illustrated in Figure 6 where one can see that the $N \log N$ implementation compares very well to the original quadratic implementation at a much reduced cost. We apply this sub-quadratic method on a simple probabilistic model for an image denoising problem. The aim of this example is to show that the method can be applied to larger graphs and still provide good results. The model underlined is chosen to showcase the flexibility and applicability of our method in particular when the edge-potential is non-integrable. It is not claimed to be an optimal approach to image denoising.[3] The node and edge potentials are defined as follows:

$$\begin{cases} \psi_u(x_u) & = & \mathcal{N}(x_u - y_u; 0, 0.1) \\ \psi_{uv}(x_u, x_v) & = & \mathcal{L}^\lambda(x_u - x_v; 0, 0.03) \end{cases} , \tag{16}$$

where $\mathcal{L}^\lambda(x; \mu, \beta) = \mathcal{L}(x; \mu, \beta)$ if $|x| \leq \lambda$ and $\mathcal{L}(\lambda; \mu, \beta)$ otherwise. In this example we set $\lambda = 0.2$. The value assigned to each pixel of the reconstruction is the estimated mean obtained over the corresponding node (figure 7). The image has size $50 \times 50$ and the simulation was run with $N = 30$ particles per nodes, $M = 5$ and 10 BP iterations taking under 2 minutes to complete. We compare it with the result obtained with EP on the same model.

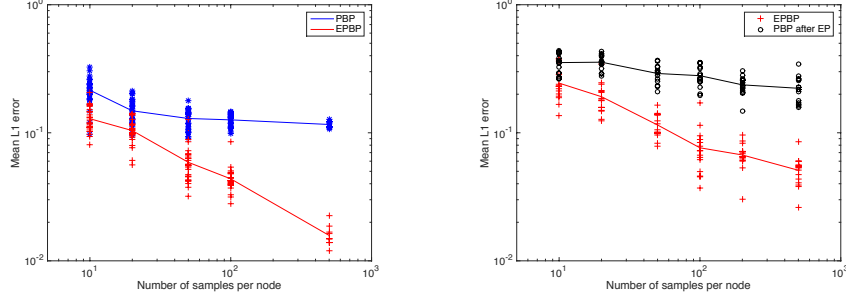

Figure 2: (left) Comparison of the mean $L^1$ error for PBP and EPBP for the $3 \times 3$ grid example. (right) Comparison of the mean $L^1$ error for "PBP after EP" and EPBP for the tree example. In both cases, EPBP is more accurate for the same number of samples.

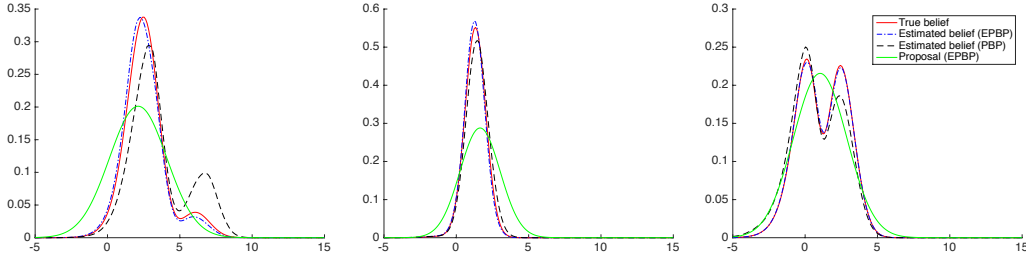

Figure 3: Comparison of the beliefs on node 1, 5 and 9 as obtained by evaluating LBP on a deterministic mesh (*true belief*), with PBP and with EPBP for the $3 \times 3$ grid example. The proposal used by EPBP at the last step is also illustrated. The results are obtained with $N = 100$ samples on each node and 20 BP iterations. One can observe visually that EPBP outperforms PBP.

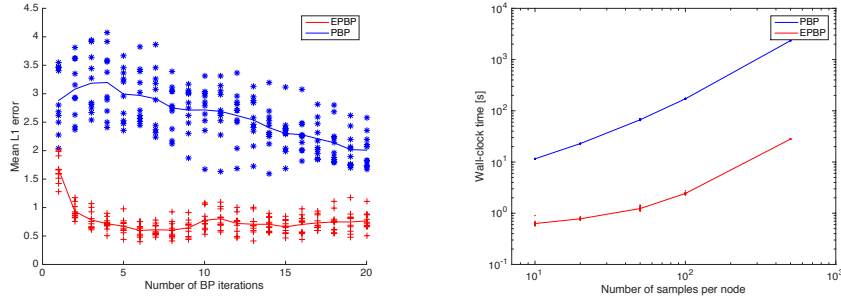

Figure 4: (left) Comparison of the convergence in $L^1$ error with increasing number of BP iterations for the $3 \times 3$ grid example when using $N = 30$ particles. (right) Comparison of the wall-clock time needed to perform PBP and EPBP on the $3 \times 3$ grid example.

## 5  Discussion

We have presented an original way to design adaptively efficient and easy-to-sample-from proposals for a particle implementation of Loopy Belief Propagation. Our proposal is inspired by the Expectation Propagation framework.

We have demonstrated empirically that the resulting algorithm is significantly faster and more accurate than an implementation of PBP using the estimated beliefs as proposals and sampling from them using MCMC as proposed in [1]. It is also more accurate than EP due to the nonparametric nature of the messages and offers consistent estimators of the LBP messages. A sub-quadratic version of the method was also outlined and shown to perform almost as well as the original method on

mildly multi-modal models, it was also applied successfully in a simple image denoising example illustrating that the method can be applied on graphical models with several hundred nodes.

We believe that our method could be applied successfully to a wide range of applications such as smoothing for Hidden Markov Models [18], tracking or computer vision [19, 20]. In future work, we will look at considering other divergences than the KL and the "Power EP" framework [21], we will also look at encapsulating the present algorithm within a sequential Monte Carlo framework and the recent work of Naesseth et al. [22].

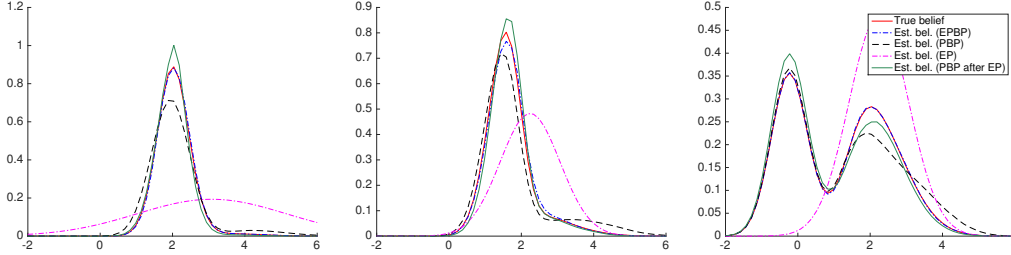

Figure 5: Comparison of the beliefs on node 1, 3 and 8 as obtained by evaluating LBP on a deterministic mesh, using EPBP, PBP, EP and PBP using the results of EP as proposals. This is for the tree example with $N = 100$ samples on each node and 20 LBP iterations. Again, one can observe visually that EPBP outperforms the other methods.

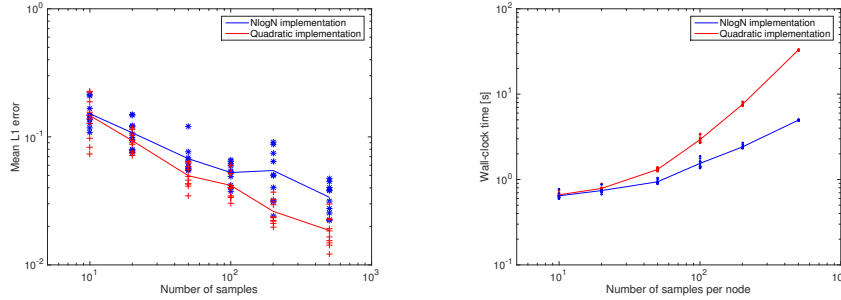

Figure 6: Comparison of the mean $L^1$ error for PBP and EPBP on a $3 \times 3$ grid (left). For the same number of samples, EPBP is more accurate. It is also faster by about two orders of magnitude (right). The simulations were run several times for the same observations to illustrate the variability of the results.

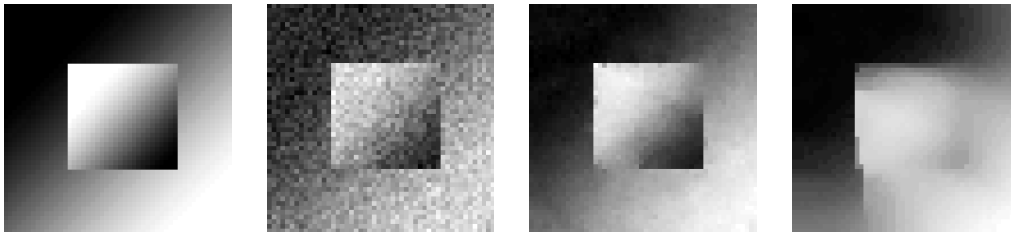

Figure 7: From left to right: comparison of the original (first), noisy (second) and recovered image using the sub-quadratic implementation of EPBP (third) and with EP (fourth).

**Acknowledgments**

We thank Alexander Ihler and Drew Frank for sharing their implementation of Particle Belief Propagation. TL gratefully acknowledges funding from EPSRC (grant 1379622) and the Scatcherd European scholarship scheme. YWT's research leading to these results has received funding from EPSRC (grant EP/K009362/1) and ERC under the EU's FP7 Programme (grant agreement no. 617411). AD's research was supported by the EPSRC (grant EP/K000276/1, EP/K009850/1) and by AFOSR/AOARD (grant AOARD-144042).

## Footnotes

[1] The scheduling used alternates between the classical orderings: top-down-left-right, left-right-top-down, down-up-right-left and right-left-down-up. One "LBP iteration" implies that all nodes have been updated once.

[2]https://github.com/tlienart/EPBP.

[3]In this case in particular, an optimization-based method such as [17] is likely to yield better results.

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
