[Reviews · NeurIPS 2015]

Submitted by Assigned_Reviewer_1

This paper introduces a new importance sampling distribution for Loopy Belief Propagation, building upon several other contributions. I could not spot any issues with the detail presented in the paper. I believe it that would have benefited from a more detailed discussion of potential applications, and perhaps a more interesting application presented in the experiments section, to draw wider interest. The significance of the paper is the main question mark in my mind.
Summary: The paper builds upon and combines several novel contributions in the literature, producing a novel approach to LBP based approximation with few restrictions on the form of the potential functions. The paper is well written.

Submitted by Assigned_Reviewer_2

The authors propose a variation on particle belief propagation in which the proposal distribution is adaptively selected at each step of the algorithm by using an expectation-propagation like calculation: the particle messages are mapped to an exponential family distribution (Gaussian for the experiments).

The idea is simple but appealing.

The authors show that it is more efficient than a previously proposed MCMC procedure for resampling the particles (whose inner loop is comparatively slow and requires user selection or tuning), and a "fixed" proposal based on EP which is not updated during the particle message passing.

The experiments are sufficient to support the claims but could have been stronger.

The authors assess the convergence behavior of the algorithm in very simple models (few variables; mildly multimodal or skewed potentials; amenable to simple discretization for comparison), with a simple synthetic denoising example as well.

While sufficient to demonstrate the point of the paper, that the EP-like proposals improve convergence in the number of samples, a practical application or example would have strengthened the submission.

The authors also mention a practical speed-up based on Briers et al. (2010), which uses subsampling to reduce the computational overhead.

While potentially useful, I did not see a significant contribution here.

Also, its usefulness and appropriateness will likely depend on the model and number of samples being used.

Given that fact, the simple empirical assessment was not strongly compelling.

Overall, the paper provides a simple but appealing idea for designing proposals during particle BP.

However, it could also benefit from a stronger and more detailed experimental evaluation, in more compelling or practical domains.
Summary: The paper provides a simple but appealing idea for designing proposals during particle BP.

However, it could also benefit from a stronger and more detailed experimental evaluation, in more compelling or practical domains.

Author Feedback
Author rebuttal: Thanks to all 6 reviewers for your helpful comments, we try to address all points raised in what follows:

To Reviewer 1,
* thank you for the positive feedback, as for your last comment regarding theoretical guarantees, this is something we are currently looking into considering, among other things, "convergent EP" of Heskes and Zoeter, 2002.

To Reviewer 2,
* re: "better explanation of how to find the proposal" the proposal at each node at a given step is the current approximation of the belief on that node. I.e.: the product of the approximated messages coming into that node, this is explained in line 196-209 (page 4), in particular equation 14 gives the form of the proposals. This is one of the main point of our submission i.e., suggesting to construct a proposal on the go using EP. We could try to make sure this point is clearer.
* re: the paper by Eslami et al., thanks for the reference, the way they approximate messages indeed bears similarity with our method but the aim of their paper (learning the message map) seems to be quite distinct. We will look into it further and add it to the references.

To Reviewer 4 with respect to the speed-up
* we do not claim that this speed-up is a significant contribution of this paper (indeed it was presented as is in Briers, 2010 and also "Sequential Auxiliary Particle Belief Propagation" [Briers et al, 2005]). Rather, we suggest that it may be a potentially useful tool which we believe can work well in this case and bring the complexity to sub-quadratic. It could also be used in PBP but their algorithm would remain quadratic due to the complexity of their sampling step.
We take your point that the empirical assessment of it being appropriate was not sufficient and will look into a more careful use of that tool in further examples.

To Reviewers 4-5 and 6 with respect to potential applications,
* to the best of our knowledge, inference on continuous MRF has numerous applications (+) and the two existing "standard" methods for going about it are NBP and PBP. (Note that the experiments they use in their paper are of comparable complexity to ours with simpler potentials). Our aim was to show that one could do better with a more automated method. NBP and PBP are both computationally expensive and NBP (which is quite widely spread) cannot deal with non-normalizable potentials, is expensive and requires KDE. PBP can work well but ideally requires sampling from the estimated beliefs which is hard, their MCMC approach is expensive and introduces a bias. This is where we believe our contribution is useful by making PBP consistent and automating the construction of the proposal.
(+) Tracking and Sensor Localization [Sudderth et al 04, Han et al 06], Stereo Vision [Sun et al 05, Klaus et al 06, Yang et al 06], Optical flow estimation [Noorsham and Wainwright 2013], Protein Folding [Peng et al], ...

Additionally, we are currently looking into the specific problem of smoothing on Hidden Markov Models (chain MRF) for nonlinear nongaussian models which also appears in many places (Signal Processing, Computer Vision,...) but is currently hard to do (Forward Filtering Backward Smoothing tends to not work well with nongaussian models in our experience and Generalized Two Filter Smoothing requires the selection of a pseudo prior which is not easy in general but where our method could play a role).
(cf "Smoothing algorithms for state-space models", Briers Doucet and Maskell, 2010)

To conclude, we will work to add a better description of the potential applications and relevance of the algorithm in the submission. We also take the point that the experiments could have been stronger and will keep working towards larger, more realistic and hopefully more convincing experiments.

Again, many thanks for your comments,
Best,
The authors